# Evaluation of Two Low-Cost Optical Particle Counters for the Measurement of Ambient Aerosol Scattering Coefficient and Ångström Exponent

**DOI:** 10.3390/s20092617

**Published:** 2020-05-04

**Authors:** Krzysztof M. Markowicz, Michał T. Chiliński

**Affiliations:** 1Institute of Geophysics, Faculty of Physics, University of Warsaw, Pastuera 5, 02093 Warsaw, Poland; 2Faculty of Biology, University of Warsaw, Miecznikowa 1, 02096 Warsaw, Poland; mich@igf.fuw.edu.pl

**Keywords:** aerosol, low-cost sensors, nephelometer, scattering coefficient, Ångström exponent

## Abstract

The aerosol scattering coefficient and Ångström exponent (AE) are important parameters in the understanding of aerosol optical properties and aerosol direct effect. These parameters are usually measured by a nephelometer network which is under-represented geographically; however, a rapid growth of air-pollution monitoring, using low-cost particle sensors, may extend observation networks. This paper presents the results of co-located measurements of aerosol optical properties, such as the aerosol scattering coefficient and the scattering AE, using low-cost sensors and using a scientific-grade polar Aurora 4000 nephelometer. A high Pearson correlation coefficient (0.94–0.96) between the low-cost particulate matter (PM) mass concentration and the aerosol scattering coefficient was found. For the PM_10_ mass concentration, the aerosol scattering coefficient relation is linear for the Dfrobot SEN0177 sensor and non-linear for the Alphasense OPC-N2 device. After regression analyses, both low-cost instruments provided the aerosol scattering coefficient with a similar mean square error difference (RMSE) of about 20 Mm^−1^, which corresponds to about 27% of the mean aerosol scattering coefficient. The relative uncertainty is independent of the pollution level. In addition, the ratio of aerosol number concentration between different bins showed a significant statistical (95% of confidence level) correlation with the scattering AE. For the SEN0177, the ratio of the particle number in bin 1 (radius of 0.15–0.25 µm) to bin 4 (radius of 1.25–2.5 µm) was a linear function of the scattering AE, with a Pearson correlation coefficient of 0.74. In the case of OPC-N2, the best correlation (r = 0.66) was found for the ratio between bin 1 (radius of 0.19–0.27 µm) and bin 2 (radius of 0.27–0.39 µm). Comparisons of an estimated scattering AE from a low-cost sensor with Aurora 4000 are given with the RMSE of 0.23–0.24, which corresponds to 16–19%. In addition, a three-year (2016–2019) observation by SEN0177 indicates that this sensor can be used to determine an annual cycle as well as a short-term variability.

## 1. Introduction

Atmospheric aerosol is of global concern due to its detrimental health effects as well as climate effects [1,2]. Monitoring ambient particles for air quality is mostly focused on ground-level mass concentrations [3], while in climate research it is focused on columnar and vertical profiles of optical properties. Various studies have reported that the relationship between columnar and surface aerosol microphysical properties is not a straightforward problem [4], which can be explained by complex (multi layers) vertical variability of aerosol chemical composition and size distribution. Therefore, the relationship between aerosol optical depth (AOD) and the particulate matter (PM) mass concentration at the ground-level should be determined regionally, to account for its specific conditions [5]. Bennouna et al. [4] reported a moderate correlation (r = 0.58) between daily mean PM_10_ mass concentration and AOD in North-Central Spain, whereas the correlation increases for monthly (r = 0.74) and yearly (r = 0.89) means. Szczepanik and Markowicz [6] showed a negative correlation coefficient between AOD and the surface aerosol scattering coefficient in Poland, which is the result of the seasonal variability of anthropogenic emission and long-range transport of natural aerosol. However, the correlation coefficient defined for instantaneous data increases to 0.48 during inversion conditions and to 0.89 under convective conditions. The empirical relation between AOD and PM mass concentration measurements for different parts of the world shows a wide range of correlation coefficients (e.g., [7,8,9,10]), indicating an important role of the variations within local meteorological conditions, the occurrence of multiple aerosol layers, and variations within aerosol chemical composition [5,9].

Particle matter (PM) mass concentration is usually measured using a relatively small number of devices, and therefore the density of network stations is often unsatisfactory, especially in the urban setting, where a large spatial and temporal inhomogeneity of emission sources and airborne particle concentrations are observed. Therefore, human exposure to aerosol particles, particularly of the fine fraction, is difficult to determine [11] and uncertain [12]. Thus, the development and implementation of portable sensors is needed to better constrain and understand human exposure to airborne pollution sources [13]. In aerosol climate research, the density of stations is even lower. However, in this case, the spatial and temporal variability of aerosol columnar optical properties (which are used to estimate aerosol radiative forcing) is lower than the variability of PM mass concentration [14]. Gaps in the spatial coverage of monitoring stations are reduced by satellite observations [15,16,17]. Unfortunately, only aerosol optical depth (AOD) has acceptable spatial coverage and is of a relatively good quality [18]. In the case of the PM mass concentration, data quality and representativeness are much worse and thus more problematic for data assimilation in aerosol transport models [16,19].

Therefore, during the last decade, there has been a rapid growth in the development of simple (inexpensive) sensors for PM mass concentration monitoring [20,21,22,23]. However, understanding the uncertainty of the aerosol number concentration measured by such devices allows an accurate enough determination of the PM mass concentration at high spatial resolutions [24]. In addition, these low-cost air quality sensors have several artefacts. For example, water uptake leads to the overestimation of the PM mass concentration at high relative humidity (RH), while gaps of air flow control or flow speed measurements produce additional uncertainties [3,25,26,27,28]. Previous research [3,27] shows that the effect of particle growth with RH can be corrected assuming physically reasonable κ values (κ-Köhler theory). Di Antonio et al. [27] report a reduction in the PM mass concentration overestimation from the factor of 5 before correction, to 1.05 after correction. Crilley et al. [3] showed that the application of the RH correction resulted in the OPC-N2 measurements being in 20–33% in agreement with the reference optical particle counter. In addition, they reported on inter-unit precision for the 14 OPC-N2 sensors of 22%  ±  13% for the PM_10_ mass concentration. Low-cost sensors are used not only for monitoring the air quality at the Earth’s surface, but also for profiling the low troposphere. These sensors are mounted on unmanned aerial vehicles (UAV; [29,30]), tethered balloons [31,32,33,34], and cable cars [35] and provide information on the microphysical and optical vertical structure of aerosol in PBL. In addition, Markowicz et al. [34] developed a method for retrieving the profiles of aerosol optical properties (single-scattering albedo) on the basis of a combination of soundings made with a miniaturized aethalometer (to measure the absorption coefficient) mounted on a tethered balloon and Raman lidar (to measure the aerosol extinction coefficient) profiling. Chilinski et al. [36] used a UAV with miniaturized devices together with ground-based lidar measurements for the estimation of absorbing AOD. However, both methods are limited to the vertical range between lidar overlap-free altitude (usually several hundred meters) and the maximum range of the tethered balloons or UAV soundings (approx. 1–2 km). The geometric overlap in the lidar system makes it impossible observation of surface haze developing during night temperature inversions. Therefore, the motivation for this study was to evaluate low-cost micro particle counters for the measurement of the ambient aerosol scattering coefficient and the scattering Ångström exponent (AE). Such devices together with AethLabs (San Francisco, CA, USA) micro-aethalometers (e.g., AE-51, MA200), can be used to measure profiles of aerosol single-scattering properties from the ground level on-board different vertical sounding platforms (UAV, balloon, cable car).

Such an opportunity results from the previous research focused on the relationship between the PM mass concentration and the aerosol scattering coefficient measured using nephelometers and professional devices for air quality monitoring. Both quantities are usually highly correlated [37,38]. For example, Carrico et al. [38] reported the determination coefficient of 0.8 in an urban (Atlanta) region. A great deal of research has been focused on the mass scattering efficiency (MSE), which is the ratio of the aerosol scattering coefficient to the PM mass concentration [39,40,41]. Cappa et al., reported in 2016 that this quantify cannot always be treated as a constant in estimating mass concentrations from scattering measurements, or vice versa. It may vary temporally and geographically [37], due to a different aerosol chemical composition and size distribution, mixing state, and age. For example, Waggoner and Weiss [42] reported an MSE for particle diameter below 2.5 µm ranging from 3 to 3.2 m^2^/g at four rural and urban stations in the Western United States, while White et al. [43] reported the same value of 2.4 at Spirit Mountain and 2.7 m^2^/g in Hillside. Chow et al. [37] reported a higher value (5.4 m^2^/g) in Mexico City. Hand and Malm [44] reported on a review of MSEs for specific aerosol species obtained from 60 studies starting in 1990. The mean MSE for fine mode dry ammonium sulfate was 2.5 ± 0.6 m^2^/g, and 2.7 ± 0.5 m^2^/g for dry ammonium nitrate. In the case of the fine mode particulate organic matter, dust, and sea salt the MSEs are 3.9 ± 1.5 m^2^/g, 3.3 ± 0.6 m^2^/g, and 4.5 ± 0.9 m^2^/g, respectively. Lower values have been reported for coarse mode dust (0.4–0.7 m^2^/g) and sea salt (0.72–1.0 m^2^/g). Such results have been estimated statistically using the multiple linear regression of compound mass concentrations on the aerosol scattering coefficient. Lowenthal and Kumar, [45] within the Interagency Monitoring of Protected Visual Environments network, have showed that MSE for PM_2.5_ increases along with rising levels of the aerosol scattering coefficient. This can be explained by the growth of the particle size distributions into size ranges with higher scattering efficiencies under more polluted conditions. In addition, their reported MSE for sulfate increased as a function of particle size over the accumulation mode (0.1–0.7 µm), while the optical efficiency increased in this size range. On the other hand, Jung et al. [46] reported a nonlinear decrease in MSE for five different aerosol chemical compositions obtained from a numerical simulation applied to polydispersed (long-normal) particles with a geometric mean diameter size range 0.5–2.5 µm.

In this study, we present a comparison of two different low-cost particle counters with the Aurora 4000 nephelometer obtained based on a long-term (over two years) co-located observation. The main goal of this research was to estimate the uncertainty of low-cost sensors with determined aerosol optical properties. This may help answer the question of whether such sensors (with micro-aethalometer) onboard a UAV or a balloon can be used to determine the profile of aerosol single-scattering albedo in the lower troposphere.

## 2. Experimental Setup

Experimental measurements were conducted at the SolarAOT research station between January 2016 and June 2018. The station is localized in Southeast Poland (49.88 °N, 21.86 °E) at 443 m a.s.l. (Figure 1). Due to its location far away from industrial and local sources of anthropogenic emissions, this site can be classified as a background (rural) station with relatively little impact of local (agriculture, transport) anthropogenic emissions. The SolarAOT station is part of the AERONET [47] and Poland-AOD (www.polandaod.pl) network [48]. The experimental setup (Figure 2) consists of a polar nephelometer and two low-cost sensors (OPC-N2 and SEN0177). In addition, for two weeks (Oct-Nov of 2018) we used two SEN0177s to estimate the statistics between both sensors at different time averaging. All instruments were connected to an inlet mounted about 3.5 m above ground level (1 m above the container roof). The aerosol pipe was heated inside the container (Figure 2) to keep the temperature of air about 10 °C higher than ambient (outside temperature). Due to the heater and additional energy dissipated by the electronics, laser diode, etc., the aerosol measurement was done in dry conditions. The temperature and RH of the measured aerosol was monitored inside the nephelometer only. The RH reported by the nephelometer was below 45% (RH < 35% for 89% of data). Thus, the hygroscopic effect is marginal and has not been taken into account in this study.

### 2.1. Nephelometer Aurora 4000

The aerosol scattering coefficient was measured by a three-wavelength polar nephelometer (Aurora 4000, Ecotech, Melbourne, Australia) [49]. The light source of this device is an array of LEDs (light-emitting diodes) with an opal glass. The electrical drive current of each LED was adjusted so that the angular intensity distribution of the light source was close to a Lambertian radiation distribution [50]. The Aurora 4000 allows the measurement of polar scattering on the basis of the following system. A motorized backscatter shutter mounted on the light source block moves periodically between several pre-programmed positions. In the backscatter position, the shutter blocks the light emitted in the forward direction. The remaining scattered light is seen by the detector as hemispheric backscattering. When the backscatter shutter is stopped at a specific angle, the radiation scattering from that angle to 170° is measured. Each observation cycle also includes a measurement without the backscatter active or a 0°-angle measurement. This allows the computation of the aerosol asymmetry parameter. During this research, the Aurora 4000 was configured to detect photons scattered in the four following sectors: 10–170°, 40–170°, 70–170°, and 90–170° at 450, 525, and 635 nm. The Hamamatsu H7155-01 Photo Multiplier Tube (PMT) detector was sampled with 10 s resolution, and the data were then averaged over 5 min. The nephelometer was calibrated with CO_2_ every 2–3 months or more often if the zero offset (Rayleigh scattering) was above 2.0 Mm^−1^. A zero-check calibration was performed every day between 00:00 and 00:30 UTC. Data processing included the following corrections: zero calibration (Rayleigh scattering), non-Lambertian illumination, and the angular truncation error. For the last correction, we used a similar method as that used by Anderson et al. [51]. However, this technique was modified by estimation of the correction factor based on the AE, as well as the ratio of backscattering to scattering coefficient (see [6]).

### 2.2. SEN0177 Optical Counter Sensor

The DFRobot (Shanghai, China) SEN0177 sensor is a low-cost sensor for real-time measurements of PM [52,53]. This sensor allows retrieval of particle mass concentration down to a minimum diameter of 0.3 µm with maximum PM_10_ mass concentration of 500 μg/m^3^ at less than 10 s response time. The SEN0177 is very compact (65 × 42 × 23 mm) and light (41 g), making it suitable for integration as a portable device that can be mounted on small balloons, UAVs, cable cars, etc. The hardware design of the PM mass concentration sensing circuit is composed of several components. Air flow inside the detector is forced by an exhaust fan, which provides probably quite a stable but not measured flow speed. The optical system (detection chamber) consists of a laser diode and a photodetector, which detects the scattered light intensity at 90°. The detector generates a current pulse proportional to the scattered light intensity. The detection cell is designed to isolate the laser from the ambient light to avoid error. Finally, the electronics with a fast analogue-to-digital converter (~1 M samples per second) and microprocessor process the light intensity to PM_1_, PM_2.5_, and PM_10_ mass concentration as well as the aerosol number concentration for particles with diameter greater than 0.3, 0.5, 1.0, 2.5, 5.0, and 10.0 µm. Data processing includes two stages. During the first stage, a Fourier transformation is applied to the light intensity measured by the detector [53]. In the second step, the inverse problem is solved, and the particle size distribution is retrieved on the basis of the frequency domain signal and Lorenz-Mie theory. Finally, the particle number concentration is converted to the mass concentration for a given aerosol density. Detailed information on numerical algorithms are not revealed by the manufacturers.

### 2.3. OPC-N2 Optical Counter Sensor

OPC-N2 is an optical aerosol sensor produced by AlphaSense (http://www.alphasense.com; Essex, UK). The OPC-N2 is slightly larger than the SEN0177 at 75 × 60 × 65 mm, and it weighs less than 105 g. The OPC-N2 samples via a small fan aspirator and classifies each particle size across 16 bin sizes (between 0.38 and 17 μm) at rates of up to ~10,000 particles per second. The flow rate is about 220 mL/min and is calculated using a time of flight method. Transit times of particles are used, and this is then used to correct the flow speed. The OPC-N2 measures the light scattered at about 658 nm by aerosol particles transported in a sample air stream (see detail description in [54]). The high frequency of light scattering measurements is used to estimate the particle size distribution via a calibration based on the Lorenz-Mie theory for spherical and uniform particles. Aerosol mass concentrations PM_1_, PM_2.5_, and PM_10_ are then calculated from the aerosol size distribution, assuming a particle density of 1.65 g/cm^3^ and refractive index of 1.5 + i0. Such an assumption, as well as the relative humidity, which is not measured inside the device (nor in the SEN0177), leads to a significant error in the PM mass concentration estimation. Details of the algorithm for conversion of the light scattering into particle number concentration and PM mass concentration are not published by AlphaSense. The number counting and sizing of the OPC-N2 was calibrated with monodispersed polystyrene latex particles by the manufacturer.

### 2.4. Data Collection and Processing

For both low-cost devices (SEN0177 and OPC-N2), as well as for the Aurora 4000, we used the serial protocol and RS232 connection to a PC computer. Data were transferred with 1 sec resolution in the case of low-cost sensors and with 10 sec in the case of the nephelometer, and then averaged to 1 min resolution. To reduce the statistical noise and fluctuation of aerosol properties, we analyzed one-hour data. In addition, 1 sec data from two SEN0177 sensors were also used to estimate the noise at a high temporal data resolution. This issue is important during profiling of the lower troposphere by miniaturized equipment onboard of a UAV, tethered balloon, cable car, and so on.

In this study, we used the aerosol scattering coefficient and scattering AE from an Aurora 4000 nephelometer. In the case of the SEN0177 and OPC-N2 the PM_1_, PM_2.5_, and PM_10_ mass concentration as well as aerosol number concentration for different bin size is analyzed. Based on nephelometer and low-cost data, we defined the regressions that are used to estimate the aerosol scattering coefficient from PM_10_ (or PM_1_, PM_2.5_) mass concentration and scattering AE from particle size distribution. The obtained aerosol optical properties from the low-cost sensors were next validated against the same Aurora 4000 nephelometer to provide the statistics between data from the low-cost devices and data from the reference instrument. This technique has some limitations but can be used, for example, when the low-cost sensors are mounted onboard a UAV, tethered balloon, or cable car. In this case, the sensor can be calibrated before or after the soundings.

## 3. Results

### 3.1. Comparison of PM Mass Concentration with Aerosol Scattering Coefficient

The one-hour data from both low-cost sensors (SEN0177, OPC-N2) as well as the Aurora 4000 were investigated in this section, and the results of data analysis are summarized in Figure 3, Table 1 and Table 2. We found that the Pearson correlation coefficient (r) for PM_1_, PM_2.5_, and PM_10_ mass concentration obtained from both low-cost sensors with the aerosol scattering coefficient (at 525 nm) is high and varies from 0.91 to 0.96. The relationship between PM_10_ (PM_1_, PM_2.5_) mass concentration and the aerosol scattering coefficient is linear for SEN0177 and nonlinear (Figure 3c) for the OPC-N2. Obtained regressions (from least squares method) are as follows:(1)$$\sigma_{\mathrm{SEN}}=5.02{\mathrm{PM}}_{10}-1.2,$$
(2)$$\sigma_{\mathrm{OPC}}=5.6{\mathrm{PM}}_{10}^{0.90},$$
where *σ*_SEN_ and *σ*_OPC_ are the aerosol scattering coefficients for SEN0177 and OPC-N2, respectively. After applying such a regression to PM_10_ mass concentration, we obtained the aerosol scattering coefficients from both low-cost devices. Figure 3b,d show the comparison of such parameters measured by the Aurora 4000 and calculated from SEN0177 and OPC-N2, respectively. Statistical parameters are similar for both low-cost sensors. The RMSE is 20.0 and 20.4 Mm^−1^ for SEN0177 and OPC-N2, respectively, which corresponds to a normalized root mean square error (NRMSE) difference of 26.9% and 27.3%. Pearson correlation coefficient between the aerosol scattering coefficient obtained from SEN0177 and OPC-N2 is 0.85, the RMSE is 33.6 Mm^−1^ and NRMSE is 50%.

We tested the same relationship between PM_10_ mass concentration and aerosol scattering coefficient for different scattering wavelengths (450 and 635 nm), but results were similar to those for 525 nm. A slightly higher Pearson correlation coefficient (r = 0.95) and lower NRMSE (21%) were obtained, while in comparison to the SEN0177 data, the light scattering (in the Aurora 4000) between 70° and 90° was used instead of the hemispheric aerosol scattering coefficient. Some improvement of the statistical relationship can be explained due to the fact that the scattering angle for the SEN0177 is close to 90°. We found, based on Aurora 4000 data, that conversion of the light scattering coefficient measured at a scattering angle between 70° and 90° to the hemispheric scattering coefficient can be performed with an NRMSE of 11%. Therefore, a part of RMSE for the Aurora 4000 and SEN0177 can be assigned to the uncertainty of the conversion of light scattering around 90° to hemispheric scattering.

In general, the relationship between the light scattering (hemispheric) coefficient and PM_10_ mass concentration is complicated. The PM_10_ mass concentration depends on aerosol size distribution and particle density, while the aerosol scattering coefficient is a function of the particle refractive index, shape, size distribution, and wavelength of light. Particle internal heterogeneity and shape are the most complex parameters to evaluate aerosol optical properties. In the case of uniform and spherical particles, the relationship between the scattering coefficient and PM_1o_ mass concentration can be determined from the Lorentz-Mie theory for. The ratio of aerosol scattering coefficient (σ) to PM_10_ mass concentration, which is the mass scattering efficiency (MSE) [44], can be obtained from
(3)$$\mathrm{MSE}\;=\frac{3}{4}\frac{\langle Q\rangle }{\rho r_{eff}}$$
where <*Q*> is the mean scattering efficiency of the cross section, ρ is the aerosol density, and *r_eff_* is the effective radius. Thus, a linear relationship between ρ and PM_10_ mass concentration corresponds to a constant value of MSE; however, MSE is a function of particle size (r) and particle chemical composition (*Q*, ρ). Therefore, the RMSE of the aerosol scattering coefficient obtained from low-cost sensors can also be related to variability of particle size distribution, shape, and chemical composition. Hence, even for very low uncertainty of PM_10_ mass concentration (e.g., measured by reference gravimetric method) and aerosol scattering coefficient, the relationship between both quantities include significant noise if the aerosol properties change. The variability of the aerosol effective radius can be described by the AE [55,56], which was measured using the Aurora 4000. We found that a reduction in the data ensemble to cases with narrow scattering AE variability, e.g., in range of 1.5–1.8 (24% of all observations), leads to a decrease in the scattering coefficient RMSE by 59% and by 57% for OPC-N2 and SEN0177, respectively. Figure 4 shows the results of systematic variability for aerosol scattering difference and AE. Such systematic variability refers to how aerosol parameters covary with each other. Analysis of the systematic relationships between aerosol optical properties is a useful metric for comparing consistency between aerosol model properties measured by different devices [57]. The systematic variability plot (Figure 4) shown here was created by binning the scattering AE from Aurora 4000 averages of the aerosol scattering coefficient difference between SEN0177 (a) or OPC-N2 (b) and Aurora 4000 into 15 bins between 0.3 and 1.2. Values above each AE bin denote the relative frequency of occurrence for each AE range. For both sensors, there is a systematic difference between scattering coefficient discrepancy and scattering AE. Up to 60% difference was found for low and high AE; however, for such cases, the number of observations is relatively low (see values above or below the bars). For 66% (SEN0177) and 50% (OPC-N2) of data, the relative difference is less than 10%. While for 97% and 90% of observation, the relative difference is lower than 20% for SEN0177 and OPC-N2, respectively.

### 3.2. Evaluation of Scattering AE from Low-Cost Sensors

Data output from both SEN0177 and OPC-N2 include information on aerosol concentration as a function of particle size. Therefore, we tested whether such data are correlated with spectral optical properties (e.g., scattering AE) measured by the Aurora 4000. For the Aurora scattering AE and micro-sensor particle number concentration ratio we found lower Pearson correlation coefficient than in PM_10_ mass concentration versus scattering coefficient, but still statistically significant. For the SEN0177 sensor, the best correlation (r = 0.74) was found for the ratio of the first (radius of 0.15–0.25 µm) and fourth (radius of 1.25–2.50 µm) bins. A high correlation (r = −0.69) was obtained for scattering AE and the effective radius was calculated based on the aerosol number concentration of five SEN0177 bins (Table 3). In the case of the OPC-N2, a statistically significant (95% of confidence level) correlation (r = 0.66) was obtained for the ratio of the first (radius of 0.19–0.27 µm) and second (radius of 0.27–0.39 µm) bins. A lower (still statistical significant at 95% of confidence level) correlation coefficient (0.38) was obtained for the first and third (radius of 0.39–0.53 µm) bins, and a negligible correlation (0.03) for the first and fourth (radius of 0.53–0.67 µm) bins. The effective radius obtained from OPC-N2 is not correlated (r = 0.02) with scattering AE.

Figure 5a,b shows a comparison of the scattering AE from Aurora and estimated from SEN0177 and OPC-N2, respectively, after log-log regression:(4)$${\mathrm{AE}}_{\mathrm{SEN}}=0.02{\left(\frac{N_{\mathrm{bin}1}^{s}}{N_{\mathrm{bin}4}^{s}}\right)}^{0.50},$$(5)$${\mathrm{AE}}_{\mathrm{OPC}}=0.33{\left(\frac{N_{\mathrm{bin}1}^{o}}{N_{\mathrm{bin}2}^{o}}\right)}^{0.59},$$
where: AE_SEN_ and AE_OPC_ are the scattering AE obtained from SEN0177; OPC-N2, *N^s^*_bin1_ (0.15–0.25 µm) and *N^s^*_bin4_ (1.25–2.50 µm) are aerosol number concentrations from SEN0177 in the first and the fourth; while *N^o^*_bin1_ (0.19–0.27 µm) and *N^o^*_bin2_ (0.27–0.39 µm) are aerosol number concentrations from OPC-N2 in the first and second bins. The RMSE is 0.23 for SEN and 0.24 for OPC, which corresponds to an NRMSE of 16% and 19% for SEN and OPC, respectively. The linear regression for the same data provides similar RMSE (Table 1 and Table 2) for scattering AE. The agreement of scattering AE obtained for both low-cost sensors with Aurora 4000 is reasonable.

### 3.3. Low-Cost Sensor Characteristics

Figure 6 shows the probability density function (pdf) for the aerosol scattering coefficient (a) and scattering AE (b) obtained from the Aurora 4000 (blue line), SEN0177 (red line), and OPC-N2 (black line). The data spread for scattering coefficient is similar for all datasets and slightly larger for the low-cost sensor than for Aurora 4000. The standard deviation is 49.8 Mm^−1^ for Aurora, 51.4 Mm^−1^ for SEN0177, and 55.8 Mm^−1^ for OPC-N2. We found the maximum of the pdf (about 30–40 Mm^−1^) for OPC-N2 is twice the SEN0177 value. In the case of the scattering AE, the low-cost sensor shows higher standard deviation (0.29 for SEN and 0.24 for OPC) in comparison to Aurora 4000 (0.14). However, the pdf shape is similar for Aurora and OPC, while for SEN, two modes (max. ~1.0 and ~1.4) are visible.

Figure 7 shows the normalized root mean square error (NRMSE) for the aerosol scattering coefficient at 525 nm (blue line) and scattering AE (red line) difference (between two SEN0177 sensors) as a function of the time averaging. The presented results were obtained over two weeks (Oct-Nov 2018) with the use of two SEN0177 sensors. In the case of the one-second data sampling (without smoothing) the NRMSE is about 17% and 7% for AE and scattering coefficient, respectively. While for 10 min the averaging is approximately 50% lower. The NMSE is below 8% and 4% for AE and scattering coefficient, respectively. On the basis of the shape of the NRMSE curves (Figure 7), we can conclude that the optimum averaging time is about one minute if fast temporal variability is needed (e.g., UAV or balloon profiling). The reduction in noise (due to data averaging) for this kind of sensor is not as high as for micro-aethalometers [34]. For the AE-51, the noise is almost 40 times lower when the data are filtered with a 10-min window [34]. This can be explained by the different measurement technique, which for AE-51 is based on the signal (filter attenuation) time derivative, and therefore the noise is sensitive to the time resolution.

### 3.4. Estimation of Aerosol Temporal Variability from Low-Cost Sensors

The low-cost sensors were tested for usability in terms of temporal variability of aerosol scattering properties. Figure 8 shows the one-hour mean aerosol scattering coefficient at 525 nm (a) and scattering AE (b) obtained from the Aurora 4000 (black dots), SEN0177 (red dots), and OPC-N2 (blue dots) during the first half of 2018. For both scattering coefficient and AE, long- and short-term variability usually matched well with the Aurora 4000 changes. For example, an increase in aerosol scattering coefficient in February was identified by all devices as well as many short-term episodes in April, May, and June. However, some discrepancy is also visible, especially for the AE. A systematic difference was seen between SEN and OPC AE in February and March.

Figure 9 shows monthly means of aerosol scattering coefficient (a) at 525 nm and scattering AE (b) obtained from the Aurora 4000 (blue bars) and SEN0177 (red bars) measurements between 2016 and 2019. The SEN0177 sensor was calibrated against the Aurora 4000 on the basis of the 2016 data. The annual cycle obtained from the SEN0177 shows a similar cycle as the nephelometer. The monthly mean aerosol scattering coefficient and AE from SEN0177 slightly overestimated the Aurora 4000 data by 3.5 Mm^−1^ and 0.05, respectively. The RMSE is 6.5 Mm^−1^ for the scattering coefficient and 0.12 for the AE, which corresponds to 9–10%.

## 4. Summary

The Alphasense OPC-N2 and DFROBOT SEN0177 sensors were evaluated for use in measurements of aerosol optical properties with an Ecotech Aurora 4000 nephelometer employed as a reference instrument. The comparison of the OPC-N2 and SEN0177 to the professional optical instrument demonstrated reasonable agreement for the low-cost sensors for the measured aerosol scattering coefficient. This quantity is calculated from linear (SEN0177) and log-log (OPC-N2) regression applied to the PM_10_ mass concentration. RMSE for scattering coefficient is 20.0 and 20.4 Mm^−1^, respectively, for OPC-N2 and SEN0177. These values correspond to a relative difference (NRMSE) of 27% for both sensors. Note that the OPC-N2 includes a mass flow sensor, which allows air flow control, while the SEN0177 works without a flow sensor, and therefore any fluctuation or long-term trends in flow speed cannot be applied in the retrieval algorithm. However, the RMSE is similar for both devices and therefore stabilization of air flow is not responsible for increased RMSE. Nonetheless, the limitation of 27% uncertainty for the aerosol scattering coefficient for SEN0177 is still acceptable for various applications. Estimating the scattering coefficient from the integrated mass distribution (PM_1_, PM_2.5_ or PM_10_) is more uncertain than calculating it from the particle size distribution. However, even for the sixteen bin size OPC-N2 device, the particle size distribution is not measured below 0.35 µm of the diameter. Thus, the calculation (e.g., based on the Lorenz–Mie theory) of aerosol optical properties from low-cost sensors particle size distribution is limited, and extensive optical properties are underestimated.

We found that for both types of sensors, the best agreement for aerosol scattering coefficient is observed when the scattering AE is between 1.2 and 1.4. This range corresponds to about 39% of the number of observations. The largest uncertainty corresponds to low (<0.5) AE or high (>1.9) AE. The first case corresponds to large (often non-spherical) particles with relatively low MSE, while the second corresponds to fine aerosol with relatively high MSE.

Previous research focusing on the validation of low-cost sensors for PM_10_ and PM_2.5_ mass concentration [26,58,59] showed that some low-cost sensors are most suited to polluted ambient environments (e.g., Shinyei PPD20V) and some to clean conditions (e.g., Shinyei PPD60V). This can be explained by the nonlinearity effect, which is reported [26,58] for some sensors (e. PMS3003, Shinyei PPD60PV) at a high value of PM mass concentration. In the case of OPC-N2, the effect on nonlinearity can be reduced during sensor calibration against a nephelometer. Karagulian et al. [59], in their review of low-cost sensors, reported that low-level devices can be used to monitor the PM level with relatively good agreement with professional devices represented by a coefficient of determination above 0.75 and a slope close to 1.0. In the case of SEN0177, the coefficient of determination is above 0.83, and in the case of OPC-N2, it is above 0.9.

Crilley et al. [3] reported that the agreement in PM_10_ mass concentration obtained from OPC-N2 in respect to reference data was 33% and 52% of the TEOM (tapered element oscillating microbalance) and GRIMM portable aerosol spectrophotometer commercial devices, respectively. Such results were obtained after taking into account the water uptake correction. Zheng et al. [26] reported that in the case of the PMS3003, the empirical nonlinear RH correction applied to PM_2.5_ suggests that this sensor can measure PM mass concentrations within ∼10% of ambient values. The agreement between the low-cost sensors and the reference one is much worse for uncorrected data, especially for high relative humidity (>90%) [27]. In the case of the aerosol scattering coefficient, the effect of relative humidity is usually not important, while nephelometers usually measure the aerosol optical properties in dry conditions. For this purpose, the air must be heated to reduce the RH below 30–40%. However, to match the lidar measurement, observation of the aerosol scattering coefficient at ambient condition is also needed. In this case, additional study is needed to establish the role of internal heating (e.g., electronic, light source) in the reduction of RH inside the detector. Unfortunately, many of them do not have temperature or RH sensors. The new version of the OPC-N3, available since 2019 [60], includes both sensors. Preliminary results show that the inside temperature is about 5–7 °C higher than ambient and the RH is below 65–70% in foggy conditions.

Comparison of the scattering AE calculated from both low-cost sensors and measured by Aurora 4000 shows also reasonable agreement. Such a quantity was computed from the particle number concentration ratio. Similar statistical parameters for scattering AE were obtained when a linear or log-log relationship was assumed. We found that the uncertainty of the scattering AE obtained after a log-log fit applied to the ratio of particle number concentration at different size is about 0.23–0.24, which corresponds to 16–19%. However, similar uncertainty was obtained from different bin ratios of aerosol number concentrations.

We found, on the basis of a data analysis of the SEN0177 sensors, that the NRMSE difference between the two devices is about 7% for the scattering coefficient and 17% for scattering AE at 1 s resolutions. The achieved results justify the use of such devices onboard tethered balloons, UAVs, or cable cars to measure the vertical profiles of aerosol optical properties in ambient conditions. The measurements close to the surface can be used by the lidar community to extend the remote sensing data from overlap to ground level and for improvement of overlap correction procedures, together with validation of algorithms used to extend lidar aerosol profiles to the Earth’s surface. The profiling of aerosol optical properties in the PBL is important during night conditions, when the vertical variability is high as a result of the near-surface or low-troposphere temperature inversion. In addition, profiling with SEN0177 or OPC-N2 together with a micro-aethalometer (AE-51, AM200) can be used to retrieve the vertical variability of single scattering albedo on the basis of the methodology proposed by Markowicz et al. [34].

## Figures and Tables

**Figure 1 sensors-20-02617-f001:**
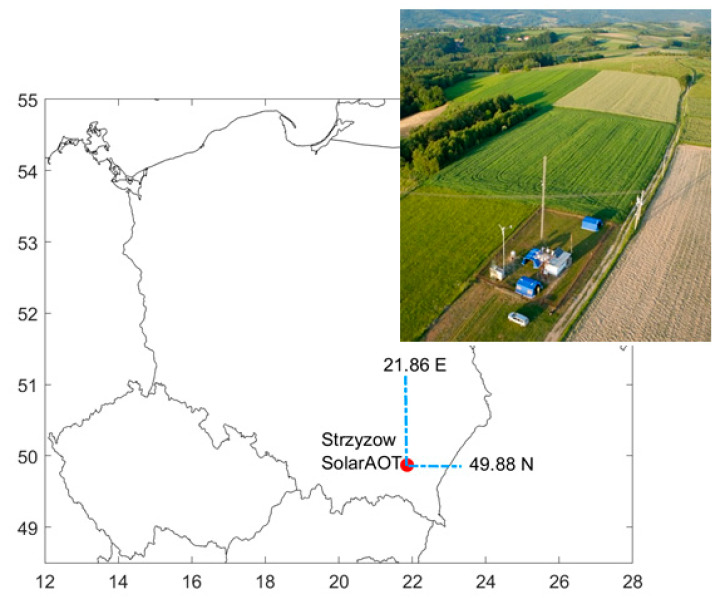
Localization of SolarAOT station in Strzyzow and site drone view.

**Figure 2 sensors-20-02617-f002:**
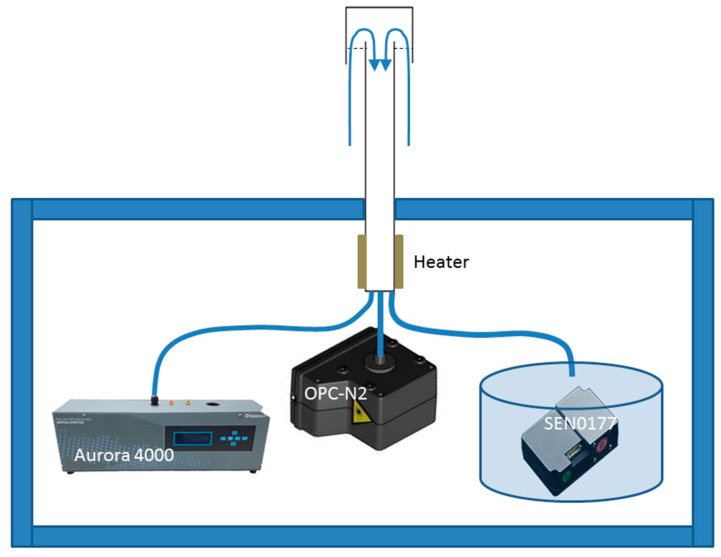
Experimental setup and connection of aerosol devices to aerosol inlet. Due to technical issues the SEN0177 sensor was mounted inside the metal box. The small fan fixed to the bottom of the box forces air flow from inlet.

**Figure 3 sensors-20-02617-f003:**
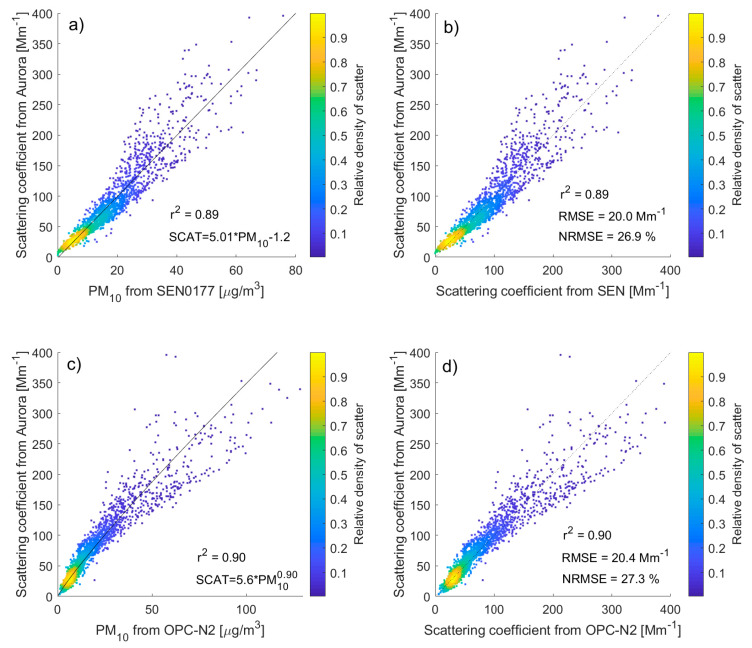
Comparison of 1-h PM_10_ mass concentration [µg/m^3^] measured by SEN0177 (**a**) and OPC-N2 (**c**) with aerosol scattering coefficient obtained from Aurora 4000 nephelometer [Mm^−1^] at 525 nm. Panels (**b**) and (**d**) show agreement between scattering coefficient measured by Aurora and obtained from regression of PM_10_ mass concentration observed by SEN0177 and OPC-N2, respectively. The solid line in panel (**a**) and (**c**) shows power law fit, and the dotted line (**b**,**d**) indicates perfect agreement. Colored dots show the density of scatter points.

**Figure 4 sensors-20-02617-f004:**
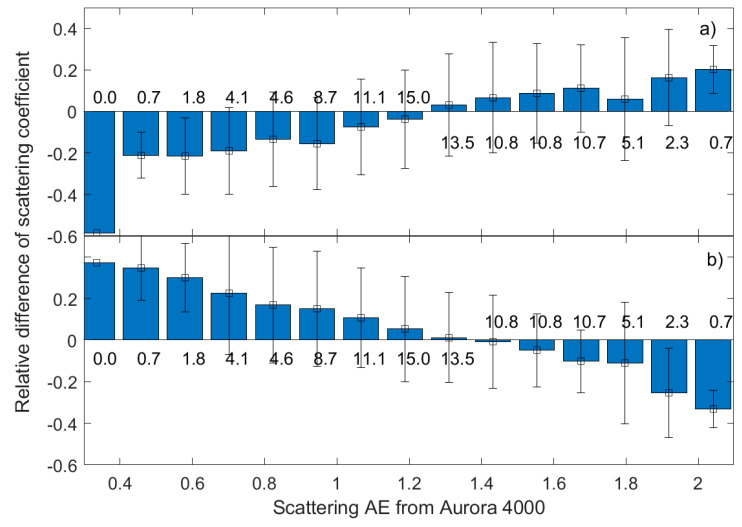
Relative difference of aerosol scattering coefficient for (**a**) SEN0177 and (**b**) OPC-N2 as a function of the scattering AE from Aurora. The error bars show standard deviations while values close to each AE bin denote the relative frequency of occurrence.

**Figure 5 sensors-20-02617-f005:**
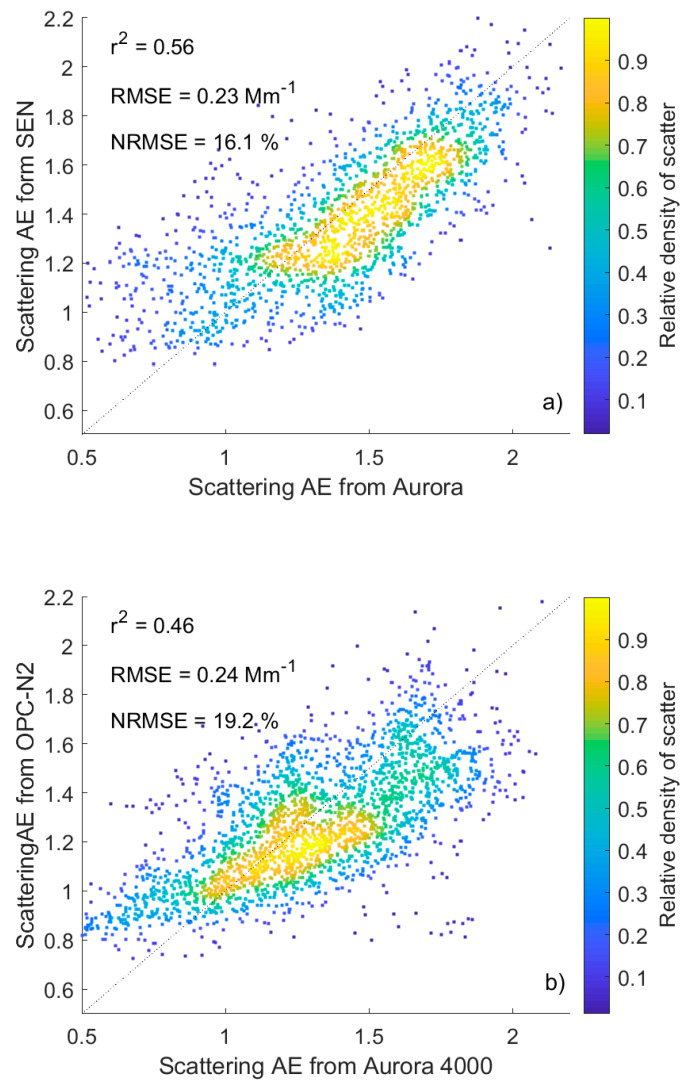
Comparison of 1-h scattering Ångström exponent calculated from SEN0177 (**a**) and OPC-N2 (**b**) and measured by Aurora 4000. The dotted line shows perfect agreement, while the colored dots indicate the scattering density.

**Figure 6 sensors-20-02617-f006:**
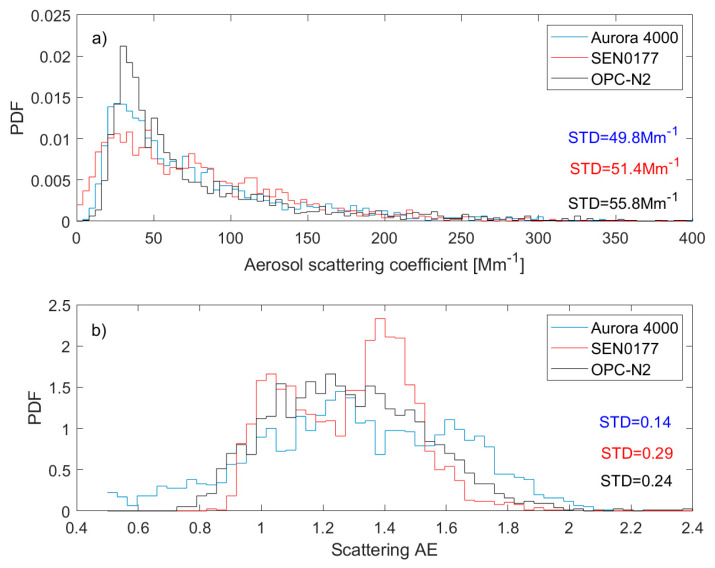
Probability distribution function for (**a**) aerosol scattering coefficient at 525 nm and (**b**) scattering AE obtained from Aurora 4000 (blue line), SEN0177 (red line), and OPC-N2 (black line).

**Figure 7 sensors-20-02617-f007:**
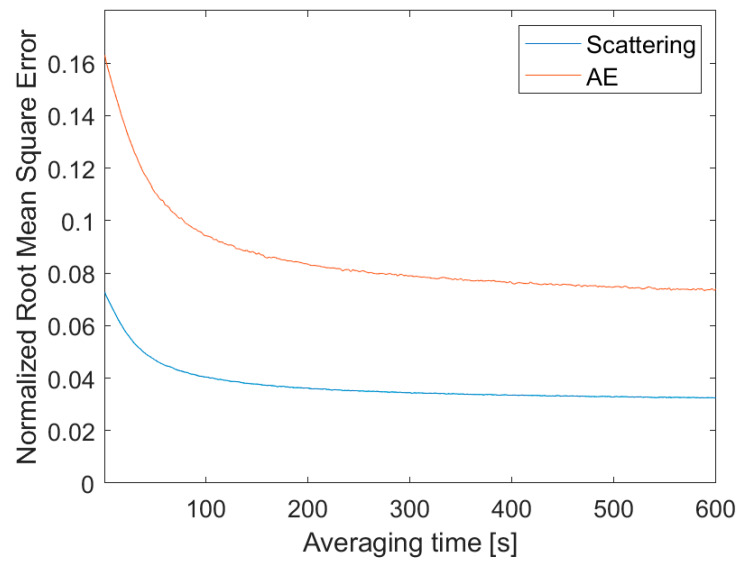
The normalised root mean square error (NRMSE) of aerosol scattering coefficient at 525 nm (blue line) and Ångström exponent (red line) difference between two SEN0177 sensors as a function of averaging time [s].

**Figure 8 sensors-20-02617-f008:**
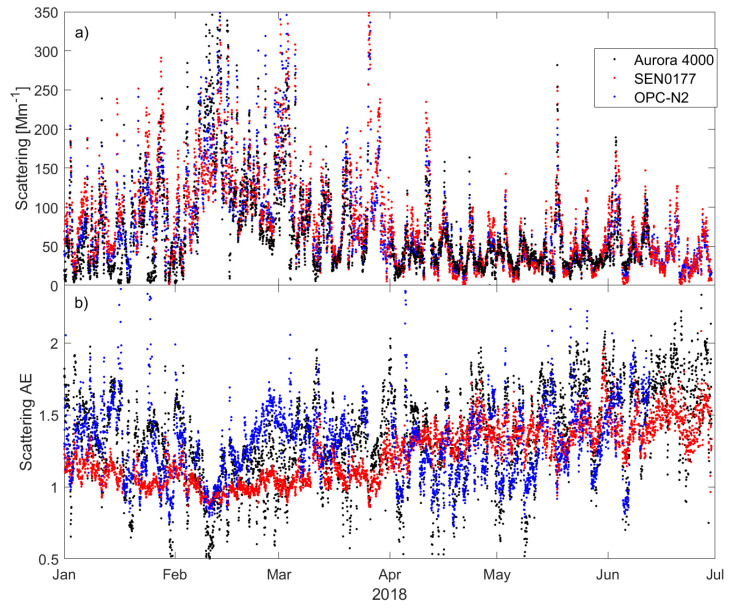
Temporal variability of (**a**) aerosol scattering coefficient at 525 nm and (**b**) scattering Ångström exponent obtained from Aurora 4000 (black), SEN0177 (red), and OPC-N2 (blue) between 1 January and 30 June of 2018.

**Figure 9 sensors-20-02617-f009:**
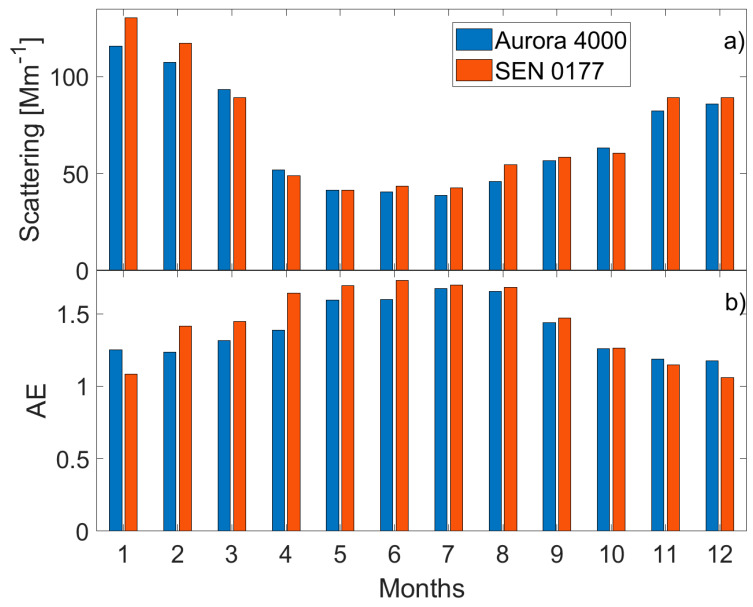
Monthly means of (**a**) aerosol scattering coefficient at 525 nm [Mm^−1^] and (**b**) scattering Ångström exponent obtained from Aurora 4000 nephelometer (blue bars) and from SEN0177 sensor based on observations performed between 2016 and 2018.

**Table 1 sensors-20-02617-t001:** Statistical parameters for comparison of SEN0177 data with Aurora 4000 aerosol optical properties in the case of linear and log-log fit. N_1_/N_4_ is the ratio of the particle number concentration between the first bin (radius of 0.15–0.25 µm) and the fourth bin (radius of 1.25–2.50 µm).

Aurora 4000	SEN 0177	Linear Fit				Log-Log Fit			
		r	Slope [Mm^−1^/µg/m^3^]	Offset [Mm^−1^]	RMSE [Mm^−1^]	r	Slope	Offset	RMSE
Scattering coefficient at 525 nm	PM1	0.91	7.71 ± 0.07	−5.1 ± 0.9	25.1	0.89	0.72 ± 0.01	2.56 ± 0.02	0.35
	PM2.5	0.93	5.70 ± 0.05	−2.5 ± 0.8	22.1	0.92	0.77 ± 0.01	2.25 ± 0.02	0.29
	PM10	0.94	5.02 ± 0.04	−1.2 ± 0.5	20.0	0.95	0.82 ± 0.01	2.06 ± 0.01	0.25
AE *	N_1_/N_4_	0.74	1.22·10^−4^ ± 3·10^−6^	0.77 ± 0.02	0.23	0.75	0.50 ± 0.01	−3.9 ± 0.1	0.23

* statistics for AE and N_1_/N_4_ are dimensionless.

**Table 2 sensors-20-02617-t002:** Statistical parameters for comparison of OPC-N2 data with Aurora 4000 aerosol optical properties in the case of linear and log-log fit. N_1_/N_2_ is the ratio of the particle number concentration between the first bin (radius of 0.19–0.27 µm) and the second bin (radius of 0.27–0.39 µm).

Aurora 4000	OPC-N2	Linear Fit				Log-Log Fit			
		r	Slope [Mm^−1^/µg/m^3^]	Offset [Mm^−1^]	RMSE [Mm^−1^]	r	Slope	Offset	RMSE
Scattering coefficient at 525 nm	PM1	0.95	3.41 ± 0.02	22.5 ± 0.5	18.2	0.98	0.75 ± 0.01	2.37 ± 0.01	0.17
	PM2.5	0.94	3.00 ± 0.02	22.2 ± 0.5	19.5	0.97	0.78 ± 0.01	2.15 ± 0.01	0.18
	PM10	0.94	3.10 ± 0.02	17.1 ± 0.6	20.1	0.95	0.89 ± 0.01	1.72 ± 0.02	0.24
	Ntot	0.96	1.42 ± 0.01	20.4 ± 0.5	16.7	0.98	0.75 ± 0.01	1.66 ± 0.02	0.16
AE *	N_1_/N_2_	0.66	0.064 ± 0.002	0.64 ± 0.02	0.25	0.68	0.59 ± 0.01	−1.12 ± 0.03	0.24

* statistics for AE and N_1_/N_2_ are dimensionless.

**Table 3 sensors-20-02617-t003:** Pearson correlation coefficient between scattering AE obtained from Aurora 4000 nephelometer and ratio of particle number concentration defined for different SEN0177 and OPC-N2 bins and effective radius.

Sensor	Bin Ratio/Effective Radius	Bin Radius Range	r	r 95% Interval
SEN0177	N_1_/N_2_	N_1_: 0.15–0.25 µm N_2_: 0.25–0.50 µm N_3_: 0.50–1.25 µm N_4_: 1.25–2.50 µm N_5_: 2.50–5.00 µm	−0.60	−0.61: −0.58
	N_1_/N_3_		0.63	0.62: 0.65
	N_1_/N_4_		0.74	0.73: 0.75
	N_1_/N_5_		0.50	0.48: 0.51
	$R_{eff}=\overset{5}{\underset{i=1}{\sum }}R_{i}^{3}N_{i/}\overset{5}{\underset{i=1}{\sum }}R_{i}^{2}N_{i}$		−0.69	−0.70: −0.68
OPC-N2	N_1_/N_2_	N_1_: 0.19–0.27 µm N_2_: 0.27–0.39 µm N_3_: 0.39–0.52 µm N_4_: 0.52–0.66 µm N_5_: 0.66–0.80 µm	0.66	0.63: 0.68
	N_1_/N_3_		0.38	0.35: 0.42
	N_1_/N_4_		0.03	0.00: 0.07
	N_1_/N_5_		−0.12	−0.21: 0.17
	$R_{eff}=\overset{15}{\underset{i=1}{\sum }}R_{i}^{3}N_{i/}\overset{15}{\underset{i=1}{\sum }}R_{i}^{2}N_{i}$		0.02	−0.01: 0.07

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
