# Peer review of "Evaluation of Two Low-Cost Optical Particle Counters for the Measurement of Ambient Aerosol Scattering Coefficient and Ångström Exponent"

_sensors, 2020, doi:10.3390/s20092617_

Round 1

Reviewer 1 Report

This work describes aerosol size distribution measurements using two mini-OPC in Poland for a two year period. These measurements are compared to the scattering coefficient and angstrom exponent from a rack mounted commercial polar nephelometer.

This work brings into light two important findings; the non-linearity of the Alphasense-N2 and the discrepancy of the calculated scattering coefficient from the mini-OPC with those measured by the nephelometer. For the latter I am not fully convinced.

Apart from these findings which would drive me to accept this article everything else is very poor. Namely

  1. Was the sample dried in all of the instruments? This is very important most of the discrepancies observed in this work may not have to do with the ambient aerosol but with the amount of water condensed on it. This important point is not discussed at all. There is a recent overview of mini OPC that stressed the importance of aerosol conditioning (Karagulian et al., 2019), while several works proposed corrections for improving this discrepancy (Manikonda et al., 2016; Di Antonio et al., 2018; Jayaratne et al., 2018; Zheng et al., 2018)
  2. The introduction is full of typos, and sentences that lack meaning or are very vague to understand their point.

What is the significant limitation that has been found?

There is no Person coefficient, it is Pearson (this mistake can be identified in the first half of the manuscript but it is corrected in the second half)

What is low-cost PM10 observation?

  1. The methods section is surprisingly poor and lacks how calculations are performed in this work. How did the authors calculate the scattering coefficient from the number size distribution? Which codes did the authors used? Is there any article related to them? What refractive index and density did the authors employ. Why was there a reason for choosing that?
  2. Was the refractive index and density constant throughout your calculations or varied based on season? Can the finding in Fig.2 be simply the outcome of poorly represented input of these values in the annual cycles? Please discuss.
  3. The scattering angstrom exponent is related to the shape of the size distribution as the authors acknowledge in Line 197. However, the authors have decided not to show any size distributions even though this is what OPC produce. Not even the size range of the mini-OPC involved is mentioned. Does the shape of the mass size distribution change during the annual cycle?
  4. How many mini instruments were used in this work? Two of each brand but how many in total to perform measurements for two years. I doubt that only one N2 operated for 2 full years. These instruments have a limited life cycle. Did the authors check the instrument-to-instrument variability using some method. This work seems to have been performed, based on Fig. 5, but it is poorly communicated. Were the instruments intercompared before deployed in the field? In Line 291 you write that . “The SEN0177 sensor was calibrated against the Aurora 4000 on the basis of the 2016 data.”. What does that mean, how was that calibration performed? Was the same procedure followed for all instruments?
  5. Comparison under ambient conditions without a section describing the annual cycle is problematic. This should be a paragraph for each of the parameter evaluated later in the text.

Less important comments that have to  be taken into account

There is a problem with the middle term in Eq. 3. What does PM10 stand for? MSE is related to the size distribution of the aerosol and its composition mainly as the right had side of Eq3 suggests. The middle term is just puzzling.

Line 129. To the best of my knowledge, there is no mass flow controller in the N2. The user can control the fan speed. This is the only parameter the

What does slightly non-linear mean? Figure 1 suggests a bi-linear system. The authors have decided to model it using power law. This is fine. There no such thing as slightly non-linear. It is a misuse of terms and should be avoided.

There seems to be a typo in Table 1 bottom of the 4th column (AE,slope of linear fit).

Lines 175-187: Most OPC include a mirror so they measure the integrated scattering within a range of angles.

The reader is not interested in bin1, bin2 or any bin from a specific instrument. He is interested in size ranges these bin represent. Please change all the text accordingly.

The fact that the best correlations between the nephelometer and the OPC is achieved for different size bins is puzzling. It most likely has to do with a combination of the inversion used by the manufacturer and the measured size distribution which involves many errors as well. But how can an author deduct any conclusion since no distributions are shown.

Did the authors measure using the both OPC for the full 2 year period? Please discuss this in the text.

Define significant, important, poor etc throughout the manuscript. What are the ranges you are using?

Define in an appendix or supplement the basic criteria you are implementing, eg RMSE.

Define abbreviations at first use

Lines 40-41 make no sense

Please add a map of the sampling location

I suggest the authors follow all the above and resubmit their article.

Karagulian, F., Barbiere, M., Kotsev, A., Spinelle, L., Gerboles, M., Lagler, F., Redon, N., Crunaire, S., Borowiak, A., (2019), Review of the Performance of Low-Cost Sensors for Air Quality Monitoring, Atmosphere, 10, 506; doi:10.3390/atmos10090506

Manikonda, A., Zíková, N., Hopke, P. K., Ferro, A. R., (2016) Laboratory assessment of low-cost PM monitors, J. Aerosol. Sci., 102, 29–40

Di Antonio, A., Popoola, O. A. M., Ouyang, B., Saffell, J. and Jones, R. L. (2018). Developing a Relative Humidity Correction for Low-Cost Sensors Measuring Ambient Particulate Matter. Sensors, 18, 2790.

Zheng, T., Bergin, M. H., Johnson, K. K., Tripathi, S. N., Shirodkar, S., Landis, M. S., Sutaria, R., Carlson, D. E., (2018), Field evaluation of low-cost particulate matter sensors in high- and low-concentration environments, Atmos. Meas. Tech., 11, 4823–4846

Jayaratne, R., Liu, X., Thai, P., Dunbabin, M., Morawska, L., (2018), The influence of humidity on the performance of a low-cost air particle mass sensor and the effect of atmospheric fog, Atmos. Meas. Tech., 11, 4883–4890.

Author Response

Authors would like to thank Reviewer for a constructive assessment of the manuscript and list of interesting question, comments and improvements.  We hope that by carefully addressing all comments made by the Reviewer we managed to improve the manuscript and make it more suitable for publication in the Journal of Sensors.

This work describes aerosol size distribution measurements using two mini-OPC in Poland for a two year period. These measurements are compared to the scattering coefficient and angstrom exponent from a rack mounted commercial polar nephelometer.

This work brings into light two important findings; the non-linearity of the Alphasense-N2 and the discrepancy of the calculated scattering coefficient from the mini-OPC with those measured by the nephelometer. For the latter I am not fully convinced.

Apart from these findings which would drive me to accept this article everything else is very poor. Namely

Re:  We have different motivation and founding than reviewer suggested.   

The motivation for this study was the evaluation of low-cost micro particle counters for the measurement of the ambient aerosol scattering coefficient and the scattering Ångström exponent (AE). Such devices together with AethLabs micro-aethalometers (e.g. AE-51, MA200) can be used to measure profiles of single-scattering properties from ground level on-board different vertical sounding platforms (UAV, balloon, cable car).

We found, that both low-cost sensors have good agreement with professional nephelometer. The achieved results justify the use of such devices on-board tethered balloons, UAVs, or cable cars to measure vertical profiles of aerosol optical properties in ambient conditions. The measurements close to the surface can be used by the lidar community to extend the remote sensing data from overlap to ground level and for improvement of overlap correction procedures, together with validation of algorithms used to extend lidar aerosol profiles to the Earth’s surface. 

We don’t know reviewer refers only to OPC-N2 sensor while we presents results for both type of sensors.

Was the sample dried in all of the instruments? This is very important most of the discrepancies observed in this work may not have to do with the ambient aerosol but with the amount of water condensed on it. This important point is not discussed at all. There is a recent overview of mini OPC that stressed the importance of aerosol conditioning (Karagulian et al., 2019), while several works proposed corrections for improving this discrepancy (Manikonda et al., 2016; Di Antonio et al., 2018; Jayaratne et al., 2018; Zheng et al., 2018)

Re: The change of relative humidity and water uptake effect is no issue in this study while the measurements were done at dry conditions (See new plot and added description in the section 2)

The introduction is full of typos, and sentences that lack meaning or are very vague to understand their point.

Re: Whole manuscript was proofreading by native speaker before submission, who check it one more time. Native speaker disagrees that introduction is full of typos.

What is the significant limitation that has been found?

Re: The largest limitation has been found when aerosol condition, described by Angstrom exponent are significant different than mean. See modified summary.

There is no Person coefficient, it is Pearson (this mistake can be identified in the first half of the manuscript but it is corrected in the second half)

Re: Corrected

What is low-cost PM10 observation?

Re: This sentence was modified

The methods section is surprisingly poor and lacks how calculations are performed in this work. How did the authors calculate the scattering coefficient from the number size distribution? Which codes did the authors used? Is there any article related to them? What refractive index and density did the authors employ. Why was there a reason for choosing that?

Re: There isn’t computation of the optical properties from aerosol number concentration in this manuscript. Such calculation does not make any sense. OPC-N2 and SEN0177 detect particles from about 0.3 um of diameter. It is well know that particles smaller than 0.3 diameter have significant contribution of optical properties such as scattering and absorption coefficient. Another way, calculation of optical properties from aerosol size distribution limited to (D>0.3 um) leads to significant underestimation of aerosol extensive parameters. In addition, very poor particle size resolution (especially in SEN0177) provides large error in calculation optical properties even if we know the refractive index.

Was the refractive index and density constant throughout your calculations or varied based on season? Can the finding in Fig.2 be simply the outcome of poorly represented input of these values in the annual cycles? Please discuss.

Re: See replay above

The scattering angstrom exponent is related to the shape of the size distribution as the authors acknowledge in Line 197. However, the authors have decided not to show any size distributions even though this is what OPC produce. Not even the size range of the mini-OPC involved is mentioned. Does the shape of the mass size distribution change during the annual cycle?

Re: This manuscript is focused on optical properties not no microphysical. Therefore we have shown data related to this parameters.

How many mini instruments were used in this work? Two of each brand but how many in total to perform measurements for two years. I doubt that only one N2 operated for 2 full years. These instruments have a limited life cycle. Did the authors check the instrument-to-instrument variability using some method. This work seems to have been performed, based on Fig. 5, but it is poorly communicated. Were the instruments intercompared before deployed in the field? In Line 291 you write that . “The SEN0177 sensor was calibrated against the Aurora 4000 on the basis of the 2016 data.”. What does that mean, how was that calibration performed? Was the same procedure followed for all instruments?

Re: Information related to these issue were added to section 2.

Comparison under ambient conditions without a section describing the annual cycle is problematic. This should be a paragraph for each of the parameter evaluated later in the text.

Re: We measured at dry condition

Less important comments that have to  be taken into account

There is a problem with the middle term in Eq. 3. What does PM10 stand for? MSE is related to the size distribution of the aerosol and its composition mainly as the right had side of Eq3 suggests. The middle term is just puzzling.

Re: We removed the middle term

Line 129. To the best of my knowledge, there is no mass flow controller in the N2. The user can control the fan speed. This is the only parameter the

Re: We contacted with manufacturer and I got information that: Flow speed is calculated using a time of flight method.  Transit times of particles are used and this is then used to correct the flow speed. We corrected the sentence in the manuscript.

What does slightly non-linear mean? Figure 1 suggests a bi-linear system. The authors have decided to model it using power law. This is fine. There no such thing as slightly non-linear. It is a misuse of terms and should be avoided.

Re: We removed the “slightly”

There seems to be a typo in Table 1 bottom of the 4th column (AE, slope of linear fit).

Re: No it isn’t.

Lines 175-187: Most OPC include a mirror so they measure the integrated scattering within a range of angles.

Re: This part of text is about the SEN0177 and Aurora 4000.

The reader is not interested in bin1, bin2 or any bin from a specific instrument. He is interested in size ranges these bin represent. Please change all the text accordingly.

Re: We added information on size range as suggested

The fact that the best correlations between the nephelometer and the OPC is achieved for different size bins is puzzling. It most likely has to do with a combination of the inversion used by the manufacturer and the measured size distribution which involves many errors as well. But how can an author deduct any conclusion since no distributions are shown.

Did the authors measure using the both OPC for the full 2 year period? Please discuss this in the text.

Re: We used only one OPC-N2.  Added such information to section 2

Define significant, important, poor etc throughout the manuscript. What are the ranges you are using?

Re: We haven’t found any papers where is the “significant, important, poor…” defined in such way?

Define in an appendix or supplement the basic criteria you are implementing, eg RMSE.

Re: We added the supplement as suggested

Define abbreviations at first use

Re: Done

Lines 40-41 make no sense

Please add a map of the sampling location

Re:  added

I suggest the authors follow all the above and resubmit their article.

Re: We added 4 papers from the list. Zheng et al., (2018) was cited in original manuscript.

Karagulian, F., Barbiere, M., Kotsev, A., Spinelle, L., Gerboles, M., Lagler, F., Redon, N., Crunaire, S., Borowiak, A., (2019), Review of the Performance of Low-Cost Sensors for Air Quality Monitoring, Atmosphere, 10, 506; doi:10.3390/atmos10090506

Manikonda, A., Zíková, N., Hopke, P. K., Ferro, A. R., (2016) Laboratory assessment of low-cost PM monitors, J. Aerosol. Sci., 102, 29–40

Di Antonio, A., Popoola, O. A. M., Ouyang, B., Saffell, J. and Jones, R. L. (2018). Developing a Relative Humidity Correction for Low-Cost Sensors Measuring Ambient Particulate Matter. Sensors, 18, 2790.

Zheng, T., Bergin, M. H., Johnson, K. K., Tripathi, S. N., Shirodkar, S., Landis, M. S., Sutaria, R., Carlson, D. E., (2018), Field evaluation of low-cost particulate matter sensors in high- and low-concentration environments, Atmos. Meas. Tech., 11, 4823–4846

Jayaratne, R., Liu, X., Thai, P., Dunbabin, M., Morawska, L., (2018), The influence of humidity on the performance of a low-cost air particle mass sensor and the effect of atmospheric fog, Atmos. Meas. Tech., 11, 4883–4890.

Reviewer 2 Report

This paper presents the use of two low cost PM sensors (the Dfrobot SEN0177 and Alphasense OPC-N2 sensors) to estimate ambient aerosol scattering coefficient and the scattering Ångström exponent. A Nephelometer Aurora 4000 was used to determine the relationships between PM concentrations and the aerosol optical properties.

The topic is in line with the interest of the Journal audience and the scope of the Sensors Journal.

However, I find this manuscript is not acceptable for publication in its current form.

The paper should be revised from a formatting, grammar, and name spelling point of view. It should be enhanced with more and clearer explanations, concerning:

  • the context/background of the study;
  • the relation with other work in the literature;
  • the objectives;
  • the methodology;
  • Results clarity.

More specifically:

  • The abstract:
    • It is difficult to understand the aim of the authors, the methodology applied and the results obtained.
  • The introduction:
    • it should include more explanations concerning the relation between columnar and surface mass concentration aerosol properties, which may be the base of the presented work.
    • Related studies in the literature should be mentioned more, particularly by highlighting the differences with the present work.
    • What are the aims of this study and what questions do you want to answer when presenting your analyses and results? This is not clear form the text.
  • A real methodology section is currently missing. Here, the only instruments are presented, but a detailed description of the whole work set-up, including the time periods when the data were collected and a list (and motivations) of the analyses produced is missing.
    • Please provide more pieces of information on the time periods adopted for your work (not clearly declared in the introduction and methodology).
    • Please explain all the analyses conducted better as they are difficult to be followed in the text. Did you use the Aurora 4000 nephelometer for both calibration and sensor validation? This is not clear form the text.
  • The results section is very difficult to follow for the reasons already mentioned. Figures may lack of labels in the panels and correct linking (fig 1) and x-y units (fig 2). Furthermore, it is not clear if the x-y axis in fig 1 should be shifted...
  • Summary:
    • I suggest the authors to expand this section by setting a discussion and a summary of their results, which may answer the questions the authors wanted to look into.
    • Some lines of the text should be included in the introduction.

Examples of specific comments (not exhaustive) are listed below. However, please refer to the points above for revision reference.

Abstract

Ln 10—13: Moving them to the introduction?Ln 14: “Coincident”. Co-located?

Ln 14: Which aerosol optical properties?

Ln 15: Person -> Pearson

Ln 16 PM observations: what kind of observations?

Ln 17: which relationship?

Ln 18: “about 27%” of what?

Ln 23, 25: bin1 e bin 4? bin1 e bin2?

Ln 27 “a-three-year observations” please specify dates better here and in the following text.

Introduction

Ln 34—35: you need to explain this better also in light of other works in the literature. (eg: https://doi.org/10.1016/j.atmosenv.2016.05.061).

Ln 38—39: This is not true! (eg: DOI: 10.1097/JOM.0000000000001277)

Ln 43—45: Please provide references.

Ln 49; “water uptake”; hygroscopicity?

Ln 50—51: “while.. uncertainties”. Please explain more and provide references.

Ln 51—53: Please explain the role of gas sensors in your work.

Ln 66—68: Please add more explanation to the aim of your study including, if it is the case, other works from the literature.

Methodology

Ln 77—78: Please specify the time periods more accurately for all the analyses conducted.

Ln 85: add reference.

Ln 97: “PMT detector” please specify.

Ln 106: add reference.

Ln 136 “.. at dry conditions” Please provide explanation.

Ln 141: How is the resampling conducted?

Ln 144: formatting

Results

Ln 146—148: “we found.. 525nm”: the sentence is not clear. Please explain

Ln 148: “all the variables”: please explain.

Ln 151: “Obtained regressions”: how are they obtained? Please explain.

Ln 151—161: This part needs to be improved. Please explain why you did you the Aurora scattering coefficients both for sensor calibrations and also validation.

Ln 160: Person.

Fig 1:

  • Labels? Missing in the panels and not clear in the caption.
  • Axes x and y do not correspond to regression equations.
  • “Panels (c) and (d) show agreement between scattering coefficient measured by Aurora and obtained from regression of PM10 observed by SEN0177 and OPC-N2, respectively”: Did you expect different results?

Ln 185: Why is it complicated?

Ln 188: Add references.

Ln 192: Which parameter?

Ln 201: please explain “systematic data analysis” and

Ln 204: “systematic variability”?

Ln 239: Why did you use one-hour measurements?

Ln 269—271: add references.

Author Response

Authors would like to thank Reviewer for a constructive assessment of the manuscript and list of interesting question, comments and improvements.  We hope that by carefully addressing all comments made by the Reviewer we managed to improve the manuscript and make it more suitable for publication in the Journal of Sensors.

This paper presents the use of two low cost PM sensors (the Dfrobot SEN0177 and Alphasense OPC-N2 sensors) to estimate ambient aerosol scattering coefficient and the scattering Ångström exponent. A Nephelometer Aurora 4000 was used to determine the relationships between PM concentrations and the aerosol optical properties. The topic is in line with the interest of the Journal audience and the scope of the Sensors Journal. However, I find this manuscript is not acceptable for publication in its current form. The paper should be revised from a formatting, grammar, and name spelling point of view. It should be enhanced with more and clearer explanations, concerning:

the context/background of the study;

the relation with other work in the literature;

the objectives;

the methodology;

Results clarity.

More specifically:

The abstract:

It is difficult to understand the aim of the authors, the methodology applied and the results obtained.

Re: We the abstract was modified as suggested

The introduction:

It should include more explanations concerning the relation between columnar and surface mass concentration aerosol properties, which may be the base of the presented work.

Re: We added a few sentences, however, this topic is not related to this study.

Related studies in the literature should be mentioned more, particularly by highlighting the differences with the present work.

Re: We added paragraph about PM versus scattering coefficient relationship

What are the aims of this study and what questions do you want to answer when presenting your analyses and results? This is not clear form the text.

Re: This issue was explained in the introduction, however, we added information about it.

A real methodology section is currently missing. Here, the only instruments are presented, but a detailed description of the whole work set-up, including the time periods when the data were collected and a list (and motivations) of the analyses produced is missing.

Re: We added such information in the section 2.

Please provide more pieces of information on the time periods adopted for your work (not clearly declared in the introduction and methodology).

Re: Added

Please explain all the analyses conducted better as they are difficult to be followed in the text. Did you use the Aurora 4000 nephelometer for both calibration and sensor validation? This is not clear from the text.

Re: Yes, I was explained in the section 2.4

The results section is very difficult to follow for the reasons already mentioned. Figures may lack of labels in the panels and correct linking (fig 1) and x-y units (fig 2). Furthermore, it is not clear if the x-y axis in fig 1 should be shifted...

Re: The Fig.1 was updated.

Summary:

I suggest the authors to expand this section by setting a discussion and a summary of their results, which may answer the questions the authors wanted to look into.

Re: The summary was modified including the information from literature

Some lines of the text should be included in the introduction.

Examples of specific comments (not exhaustive) are listed below. However, please refer to the points above for revision reference.

Abstract

Ln 10—13: Moving them to the introduction?

We removed it from manuscript

Ln 14: “Coincident”. Co-located?

Re: Done

Ln 14: Which aerosol optical properties?

Re: Added

Ln 15: Person -> Pearson

Re: Done

Ln 16 PM observations: what kind of observations?

Re: we removed the “observation”

Ln 17: which relationship?

Re: Modified to : “…PM10 – aerosol scattering coefficient relations…”

Ln 18: “about 27%” of what?

Re: added: “… of mean aerosol scattering coefficient”

Ln 23, 25: bin1 e bin 4? bin1 e bin2?

Re: Added size range for all bins

Ln 27 “a-three-year observations” please specify dates better here and in the following text.

Re: We added the years

Introduction

Ln 34—35: you need to explain this better also in light of other works in the literature. (eg: https://doi.org/10.1016/j.atmosenv.2016.05.061).

Re: We don’t why we have to start here a new tread?. The relationship between columnar and surface aerosol properties is different but of course important topic.

Ln 38—39: This is not true! (eg: DOI: 10.1097/JOM.0000000000001277)

Re: We do not agree. In the suggested by reviewer paper we can read: “Of course, individual exposure is likely to vary significantly due to circumstantial events such as occupation, time spent indoors versus outdoors, place of residency, time spent near cooking operations, even the mode of transport or route taken during a daily commute. This underscores the need for the development and implementation of portable sensors to better constrain and understand human exposure to airborne pollution sources.”

We added this citation and modified the sentence: Therefore, human exposure to aerosol particle, particularly fine fraction, is difficult to determine [4] and uncertain [5]. Thus, development and implementation of portable sensors is needed to better constrain and understand human exposure to airborne pollution sources [6].

Ln 43—45: Please provide references.

Re: We added 5 citations to these sentences.

Ln 49; “water uptake”; hygroscopicity?

Re:  Researches used both “water uptake” and “hygroscopicity” words to describe particle growths, e.g. https://www.nature.com/articles/s41597-019-0158-7

Ln 50—51: “while.. uncertainties”. Please explain more and provide references.

Re: We added information on uncertainties and a new references.

Ln 51—53: Please explain the role of gas sensors in your work.

Re: We removed this sentence

Ln 66—68: Please add more explanation to the aim of your study including, if it is the case, other works from the literature.

Re: we added a few sentence. This is the first study of using miniaturize sensors to measure aerosols scattering coefficient therefore we cannot cite any paper on that.

Methodology

Ln 77—78: Please specify the time periods more accurately for all the analyses conducted.

Re: added

Ln 85: add reference.

Re: Added reference:

Chamberlain-Ward S, Sharp F. Advances in nephelometry through the Ecotech Aurora nephelometer. ScientificWorldJournal. 2011;11:2530–2535. doi:10.1100/2011/310769

Ln 97: “PMT detector” please specify.

Re: We added:  The Hamamatsu H7155-01Photo Multiplier Tube (PMT)…

Ln 106: add reference.

Re: We added two references:

Migos,T., Christakis, I., Moutzouris, K., Stavrakas,J.: On the Evaluation of Low-Cost PM Sensors for Air Quality Estimation, 2019 8th International Conference on Modern Circuits and Systems Technologies (MOCAST), Thessaloniki, Greece, 2019, pp. 1-4.

Chee F. P.1 , Angelo S. F.1 , Kiu S. L.1 , Justin S.2 and Jackson C. H. W.: Real time particulate matter concentration measurement using laser scattering, Journal of Engineering and Applied Sciences, 13(22), 8873-8879, 2018.

Ln 136 “.. at dry conditions” Please provide explanation.

Re We removed: at dry conditions

Ln 141: How is the resampling conducted?

Re: We changed “resampled” to “averaged”

Ln 144: formatting

Re: Thanks, done

Results

Ln 146—148: “we found.. 525nm”: the sentence is not clear. Please explain

Re: We removed this sentence

Ln 148: “all the variables”: please explain.

Re: We modified this sentence.

Ln 151: “Obtained regressions”: how are they obtained? Please explain.

Re: we added least squares method

Ln 151—161: This part needs to be improved. Please explain why you did you the Aurora scattering coefficients both for sensor calibrations and also validation.

Re: We added some sentence on that in the 2.4 subsection.

Ln 160: Person.

Re: Corrected

Fig 1:

Labels? Missing in the panels and not clear in the caption.

Axes x and y do not correspond to regression equations.

“Panels (c) and (d) show agreement between scattering coefficient measured by Aurora and obtained from regression of PM10 observed by SEN0177 and OPC-N2, respectively”: Did you expect different results?

Re: The Fig.1 was corrected as suggested.

Ln 185: Why is it complicated?

We added following sentences: The PM10 depends on aerosol size distribution and particle density, while aerosol scattering coefficient is a function of particle refractive index, shape, and size distribution. Particle internal heterogeneity and shape are the most complex parameters to evaluate aerosol optical properties. In case of uniform and spherical particles relation between scattering coefficient and PM1o can be determined from Lorentz-Mie theory for.

Ln 188: Add references.

Re: We added reference

Hand, J. L., and Malm, W. C. ( 2007), Review of aerosol mass scattering efficiencies from ground‐based measurements since 1990, J. Geophys. Res., 112, D16203, doi:10.1029/2007JD008484.

Ln 192: Which parameter?

Re: We modified sentence

Ln 201: please explain “systematic data analysis” and

Re: We added two sentence with explanation

Ln 204: “systematic variability”?

Re: Added see above

Ln 239: Why did you use one-hour measurements?

Re: For the 1-hour data the instruments noise is reduced. Even for the professional Aurora 4000 the statistical noise for 1 min data is high, especially for the Angstrom exponent. For example, comparison of 1 sec data  obtained from Aurora and OPC-N2 (or SEN0177) will include significant noise corresponds also to Aurora statistic fluctuation.  

Ln 269—271: add references.

Re: Reference added Markowicz et al., 2017

Reviewer 3 Report

This paper describes results from a comprehensive measurement dataset and an appropriate analysis of that dataset. The title of the paper and the abstract provide an accurate description of the material presented (although see first minor comment below). My view is that, in general, the amount of material, its interest to the (relevant part of the) scientific community, and the quality of its presentation, are appropriate for recommendation in ‘Sensors’. I note below some minor issues, mostly ‘editorial’ in nature.

Title: Replace ‘a’ with ‘two’ so that the title begins ‘Evaluation of two low-cost optical….’

L23: State also the actual particle size ranges corresponding to bins 1 and 4.

L25: As above, state also the particle size ranges corresponding to the bin numbers.

L28: Delete spurious word ‘to’

L33: Perhaps better to write ‘ground-level’ rather than ‘surface’

L45: Insert ‘the’ before ‘last decade’

L49: Delete ‘the’ before ‘water’

L53: Replace ‘done’ with ‘undertaken’

L54: Start sentence with ‘Low-cost sensors…’

L57: Insert ‘Markowicz et al.’ before ‘[18]’

L61: Insert ‘Chilinski et al.’ before ‘[20]’

L68: Add the names of the microaeth manufacturers alongside each instrument number/name.

L72-L75: All the text from “The structure of the paper..” to the end of the section can be deleted. This text just tells the reader that the next sections are methods, results and discussion/summary, which is how every paper is structured!

L85: Correct the manufacturer name to ‘Ecotech’

L88: Change ‘a’ to ‘the’ to read ‘on the basis of the following system’

L89: Delete the inappropriate comma after ‘block’

L94: Rephrase ‘computing’ to ‘computation of’

L96: Delete ‘respectively’ – I don’t see that there is any matched ordering of items in this sentence that requires use of the word respectively.

L97: Rephrase ‘and next data was averaged over’ to ‘and the data was then averaged over’

L102: Insert ‘Anderson et al.’ before ‘[24]

L107: Change ‘up to a minimum diameter’ to ‘down to a minimum diameter’

L118: Change ‘above’ to ‘greater than’

L119: Insert ‘the’ before ‘detector’

L146: Insert ‘in Figure 1 and’ after ‘summarised’

L160: This sentence is not grammatical. Suggest starting sentence with “The Pearson….”, insert ‘the’ before ‘aerosol’, insert ‘is’ before ‘50%’, and delete ‘respectively’

L197: Insert ‘the’ before ‘AE’

L201: Only one bar colour is used in Figure 2 so can change ‘Blue bars’ to ‘The bars’

L206: Delete the word ‘blue’.

L206: Insert ‘a’ before ‘significant’

L208: The words ‘Evaluation of scattering AE from low-cost sensors’ is not a sentence. I am guessing these words should be a section sub-heading.

L280: Change start of sentence to read ‘For both scattering…’ and also insert a comma after ‘AE’

L282: Correct spelling of ‘devices’

L313: Insert ‘Crilley et al.’ before ‘[3]’

L314-315: The implication of how this sentence is currently phrased is that the TEOM is an optical counter for particles. It is not. It measures the mass of the particles by collection on an oscillating microbalances – there is no optical part to the measurement methodology. Please rephrase to avoid this implication.

L338: Insert ‘Markowicz et al.’ before ‘[18]’

L411: Add all co-author surnames and add the year of publication for the Anderson et al. reference.  

L423: Add all co-author surnames to the Noh et al. reference.

Author Response

Reviewer 3

Authors would like to thank Reviewer for a positive assessment of the manuscript and list of important correction and improvements. We hope that by carefully addressing all comments made by the Reviewer we managed to improve the manuscript and make it more suitable for publication in the Journal of Sensors.

This paper describes results from a comprehensive measurement dataset and an appropriate analysis of that dataset. The title of the paper and the abstract provide an accurate description of the material presented (although see first minor comment below). My view is that, in general, the amount of material, its interest to the (relevant part of the) scientific community, and the quality of its presentation, are appropriate for recommendation in ‘Sensors’. I note below some minor issues, mostly ‘editorial’ in nature.

Title: Replace ‘a’ with ‘two’ so that the title begins ‘Evaluation of two low-cost optical….’

Re: Done

L23: State also the actual particle size ranges corresponding to bins 1 and 4.

Re: Done

L25: As above, state also the particle size ranges corresponding to the bin numbers.

Re: Done

L28: Delete spurious word ‘to’

Re: Done 

L33: Perhaps better to write ‘ground-level’ rather than ‘surface’

Re: Done 

L45: Insert ‘the’ before ‘last decade’

Re: Done 

L49: Delete ‘the’ before ‘water’

Re: Done 

L53: Replace ‘done’ with ‘undertaken’

Re: Done 

L54: Start sentence with ‘Low-cost sensors…’

Re: Done 

L57: Insert ‘Markowicz et al.’ before ‘[18]’

Re: Done

L61: Insert ‘Chilinski et al.’ before ‘[20]’

Re: Done 

L68: Add the names of the microaeth manufacturers alongside each instrument number/name.

Re: Added AethLabs 

L72-L75: All the text from “The structure of the paper..” to the end of the section can be deleted. This text just tells the reader that the next sections are methods, results and discussion/summary, which is how every paper is structured!

Re:  We remove it

L85: Correct the manufacturer name to ‘Ecotech’

Re: Done

L88: Change ‘a’ to ‘the’ to read ‘on the basis of the following system’

Re: Done 

L89: Delete the inappropriate comma after ‘block’

Re: Done 

L94: Rephrase ‘computing’ to ‘computation of’

Re: Done 

L96: Delete ‘respectively’ – I don’t see that there is any matched ordering of items in this sentence that requires use of the word respectively.

Re: Done 

L97: Rephrase ‘and next data was averaged over’ to ‘and the data was then averaged over’

Re: Done 

L102: Insert ‘Anderson et al.’ before ‘[24]

Re: Done 

L107: Change ‘up to a minimum diameter’ to ‘down to a minimum diameter’

Re: Done 

L118: Change ‘above’ to ‘greater than’

Re: Done 

L119: Insert ‘the’ before ‘detector’

Re: Done 

L146: Insert ‘in Figure 1 and’ after ‘summarised’

Re: Done 

L160: This sentence is not grammatical. Suggest starting sentence with “The Pearson….”, insert ‘the’ before ‘aerosol’, insert ‘is’ before ‘50%’, and delete ‘respectively’

Re: Modified

L197: Insert ‘the’ before ‘AE’

Re: Done  

L201: Only one bar colour is used in Figure 2 so can change ‘Blue bars’ to ‘The bars’

Re: Done 

L206: Delete the word ‘blue’.

Re: Done 

L206: Insert ‘a’ before ‘significant’

Re: Done 

L208: The words ‘Evaluation of scattering AE from low-cost sensors’ is not a sentence. I am guessing these words should be a section sub-heading.

Re: Yes, you right, corrected.

L280: Change start of sentence to read ‘For both scattering…’ and also insert a comma after ‘AE’

Re: Done 

L292: Correct spelling of ‘devices’

Re: Done 

L313: Insert ‘Crilley et al.’ before ‘[3]’

Re: Done 

L314-315: The implication of how this sentence is currently phrased is that the TEOM is an optical counter for particles. It is not. It measures the mass of the particles by collection on an oscillating microbalances – there is no optical part to the measurement methodology. Please rephrase to avoid this implication.

Re: Thanks, this sentence was modified: “Crilley et al. [3] reported that the agreement in PM10 concentration obtained from OPC-N2 in respect to reference data was 33 and 52 % of the TEOM (tapered element oscillating microbalance ) and GRIMM portable aerosol spectrophotometer commercial devices, respectively.”

L338: Insert ‘Markowicz et al.’ before ‘[18]’

Re: Done 

L411: Add all co-author surnames and add the year of publication for the Anderson et al. reference.  

Re: Tanks added.

L423: Add all co-author surnames to the Noh et al. reference.

Re: Tanks added.

Round 2

Reviewer 1 Report

It is evident that the authors’ ability in writing English is not apt. I still support my comment that the introduction is full of sentences that make no sense. The authors should acknowledge that this is a scientific publication written in English. Proper use of English is essential

I will include in my comments also the recent additions of the authors in the revised manuscript.

Line 12: there is no “rare nephelometer network”, it can be scarce, geographically underrepresenting etc. But using the word rare is simply bad use of English.

Line 40 it is not be, it is  by. This is a typo

Line 57: human are not exposed to one particle but to particles. This is a typo

Line 57: particularly of the fine fraction or in the fine fraction. Improper use of English

Line 60-62: I really do not understand what you are talking about. Please rephrase

Line 69. Whether the uncertainty………. is enough. This means that if the uncertainty is high it will serve the purpose, if I take this sentence literally, which is the exact opposite of the message you want to deliver. This sentence should read. “However, understanding whether the uncertainty of the aerosol number concentration measured by such devices allows an accurate enough determination of the PM mass ………”. Improper English

Line 71: artefact problems. This is a  double statement, as artefact means an error and thus it is a problem. Improper use of English

After line 71 the use of English improves.

There is a miscommunication concerning the Mie calculations in this work. My impression based on Fig 3 (of the revised manuscript) was that such calculations were performed or that the authors managed to read the raw signal of both OPC. In the latter case an OPC can be used similar to a nephelometer. I apologize for the misunderstanding.

What does the 1.22±10(-4) ±3±10(-6) found on table 1 mean if this not a typo?

The authors should acknowledge that this is a scientific journal. What is significant in their view should be quantified and mentioned since they use it often in the text. Because every individual can use these terms arbitrarily. Is significant above 90%, 99% 50%? Eg what is significant noise? How is this defined. The manuscript is full of these unclear terms.

Finally the authors responded to one of my comments as found below.

Re: There isn’t computation of the optical properties from aerosol number concentration in this manuscript. Such calculation does not make any sense. OPC-N2 and SEN0177 detect particles from about 0.3 um of diameter. It is well know that particles smaller than 0.3 diameter have significant contribution of optical properties such as scattering and absorption coefficient. Another way, calculation of optical properties from aerosol size distribution limited to (D>0.3 um) leads to significant underestimation of aerosol extensive parameters. In addition, very poor particle size resolution (especially in SEN0177) provides large error in calculation optical properties even if we know the refractive index.

What they mention is true and I totally agree. However the authors should acknowledge that estimating the scattering coefficient from the integrated mass distribution, this is PM10, it is even more uncertain that calculating it from the distribution itself.

Author Response

It is evident that the authors’ ability in writing English is not apt. I still support my comment that the introduction is full of sentences that make no sense. The authors should acknowledge that this is a scientific publication written in English. Proper use of English is essential

I will include in my comments also the recent additions of the authors in the revised

manuscript.

Re:  My English is poor therefore the manuscript was twice proofreading by native speaker and ones corrected by official (sworn) translator. Therefore I don’t know why the text includes the typos and improper English sentences.

Line 12: there is no “rare nephelometer network”, it can be scarce, geographically underrepresenting etc. But using the word rare is simply bad use of English.

Re: Changed this sentence as suggested

Line 40 it is not be, it is  by. This is a typo

Re: Done

Line 57: human are not exposed to one particle but to particles. This is a typo

Re: Done

Line 57: particularly of the fine fraction or in the fine fraction. Improper use of English

Re: Changed to: particularly of the fine fraction

Line 60-62: I really do not understand what you are talking about. Please rephrase

Re: We modified this sentence.

Line 69. Whether the uncertainty………. is enough. This means that if the uncertainty is high it will serve the purpose, if I take this sentence literally, which is the exact opposite of the message you want to deliver. This sentence should read. “However, understanding whether the uncertainty of the aerosol number concentration measured by such devices allows an accurate enough determination of the PM mass ………”. Improper English

Re: Thank you, we modified this sentence as suggest

Line 71: artefact problems. This is a  double statement, as artefact means an error and thus it is a problem. Improper use of English

Re: Removed the “problem”

After line 71 the use of English improves.

Re: The whole Abstract and Introduction were proofreading.

There is a miscommunication concerning the Mie calculations in this work. My impression based on Fig 3 (of the revised manuscript) was that such calculations were performed or that the authors managed to read the raw signal of both OPC. In the latter case an OPC can be used similar to a nephelometer. I apologize for the misunderstanding.

What does the 1.22±10(-4) ±3±10(-6) found on table 1 mean if this not a typo?

Re: Yes, it was mistake, we corrected it.

The authors should acknowledge that this is a scientific journal. What is significant in their view should be quantified and mentioned since they use it often in the text. Because every individual can use these terms arbitrarily. Is significant above 90%, 99% 50%? Eg what is significant noise? How is this defined. The manuscript is full of these unclear terms.

Re: We added the confidence level to defined the statistical significant. We also modified sentence when the “significant” world was used. 

We note the “significant noise” is used only one times in the manuscript!. We used it in the general sentence: “Hence, even for very low uncertainty of PM10 mass concentration (e.g. measured by reference gravimetric method) and aerosol scattering coefficient, the relationship between both quantities include significant noise if the aerosol properties change.”

It is difficult to be more precision in such case if we don’t defined the level of aerosol optical properties changes and so on.

Finally the authors responded to one of my comments as found below.

Re: There isn’t computation of the optical properties from aerosol number concentration in this manuscript. Such calculation does not make any sense. OPC-N2 and SEN0177 detect particles from about 0.3 um of diameter. It is well know that particles smaller than 0.3 diameter have significant contribution of optical properties such as scattering and absorption coefficient. Another way, calculation of optical properties from aerosol size distribution limited to (D>0.3 um) leads to significant underestimation of aerosol extensive parameters. In addition, very poor particle size resolution (especially in SEN0177) provides large error in calculation optical properties even if we know the refractive index.

What they mention is true and I totally agree. However the authors should acknowledge that estimating the scattering coefficient from the integrated mass distribution, this is PM10, it is even more uncertain that calculating it from the distribution itself.

Re: We added the sentence on that to the Summary section.

Reviewer 2 Report

Many thanks for this revised paper, which addresses the comments of the first version.

Author Response

Thank you very much.